# Pan-cancer analysis of *B3GNT5* with potential implications for cancer immunotherapy and cancer stem cell stemness

**Feng Peng, Yechen Feng, Shuo Yu, Ruizhi He, Hebin Wang, Yu Xie\*, Renyi Qin ⓘ\***

Department of Biliary-Pancreatic Surgery, Affiliated Tongji Hospital, Tongji Medical College, Huazhong University of Science and Technology, Wuhan, China

\* ryqin@tjh.tjmu.edu.cn (RQ); YXie@hust.edu.cn (YX)

**Data Availability Statement:** All data used in this study are publicly available and described as follows: 1 TCGA data TCGA pan-cancer data was downloaded from UCSC Xena database https://toil-

## Abstract

*B3GNT5*, a critical member of the β-1,3-N-acetylglucosaminyl transferase gene family involved in lactose and glycosphingolipids biosynthesis, has been documented to promote tumor-infiltrating T-cell responses. Our research utilized the Pan-Cancer dataset from The Cancer Genome Atlas (TCGA) to explore the functional role of B3GNT5. Our study demonstrated that the antibody-driven inhibition of B3GNT5 diminished T cell-mediated anti-tumor responses in both in vitro and in vivo settings. By analyzing RNA-seq data from Genotype-Tissue Expression (GTEx) and TCGA databases, we observed differential expression levels of *B3GNT5* across various tumor types accompanied by an unfavorable prognostic correlation. We further utilized integrated clinical survival data from TCGA and immune cell infiltration scoring patterns to identify significant associations between *B3GNT5* expression and immune checkpoints, cancer stemness, chemokines, chemokine receptors, and immune-activating genes. *B3GNT5*'s expression was highly correlated with different immunoregulatory factors, including T cell infiltration, chemokine receptors, and activation genes. Subsequent experiments discovered that suppressing *B3GNT5* expression in pancreatic adenocarcinoma cells significantly reduced their tumorigenicity by limiting sphere-forming ability and self-renewal capacity, thus underscoring *B3GNT5*'s vital role as a prognostic factor in immune regulation across pan-cancer. Our findings suggest that *B3GNT5* presents a viable target for cancer immunotherapy by enabling effective communication between cancer stem cells and immune cells during tumor treatment.

## Introduction

Cancer is a major contributor to mortality rates in developed and developing countries, and the accompanying clinical load is estimated to rise in tandem with population growth and aging demographics. This phenomenon is particularly pronounced in underdeveloped nations, where approximately 82% of the worldwide populace makes its home [1]. Despite the continued progression and evolution of medical technologies, encompassing surgical procedures, radiotherapy, chemotherapy, as well as immunotherapies, the clinical outcomes for

xena-hub.s3.us-east-1.amazonaws.com/download/tcga_RSEM_gene_tpm.gz clinical data https://xenabrowser.net/datapages/?dataset=Survival_SupplementalTable_S1_20171025_xena_sp&host=https%3A%2F%2Fpancanatlas.xenahubs.net&removeHub=https%3A%2F%2Fxena.treehouse.gi.ucsc.edu%3A443 2 GTEx data GTEx data was downloaded from UCSC Xena database https://toil-xena-hub.s3.us-east-1.amazonaws.com/download/gtex_RSEM_gene_tpm.gz GTEx_phenotype https://toil-xena-hub.s3.us-east-1.amazonaws.com/download/GTEX_phenotype.gz 3 immune cell infiltration data 3.1 data from published paper https://www.cell.com/cms/10.1016/j.immuni.2018.03.023/attachment/1b63d4bc-af31-4a23-99bb-7ca23c7b4e0a/mmc2.xlsx 3.2 data from ImmuCellAI database http://bioinfo.life.hust.edu.cn/ImmuCellAI#!/resource 3.2 data from TIMER2 database http://timer.cistrome.org/infiltration_estimation_for_tcga.csv.gz 4-5 Copy number and methylation data of 33 TCGA tumors https://www.cbioportal.org/datasets

**Funding:** Feng Peng, the funders of National Natural Science Foundation of China (NO.81402443), is the first author of manuscript and has conceived and directed the study, performed data simulation and algorithm assessment and optimisation under the supervision of Renyi Qin (the funders of National Natural Science Foundation of China, NO.81772950).

**Competing interests:** The authors have declared that no competing interests exist

patients in advanced stages of cancer remain unfulfilling. This is particularly true when examining the detrimental effects that some treatments may have on these individuals [2]. Therefore, the prompt detection of molecular targets is imperative to augment the therapeutic potency and specificity. This can be effectively achieved via Pan-cancer Analysis [3].

Glycosphingolipids (GSLs) are a vast group of glycoconjugates that occur in cellular membranes. GSLs carry out distinctive functions within the cell membrane, owing to the individual core structures they possess. Compared to alternative membrane lipids, GSLs exhibit significant molecular intricacy [4]. It plays a vital role in cellular adhesion, migration, modulation of signaling proteins, and engagement with pathogens and toxins [5, 6]. *B3GNT5*, a member of the β-1,3-N-acetylglucosaminyl transferase family, catalyzes the transfer of N-acetylglucosamine (GlcNAc) from UDP-GlcNAc to galactose positioned at the non-reducing end of the carbohydrate chain through β-1,3-linkage. This enzyme plays a vital role in lactose biosynthesis and generates new lactose series of glycosphingolipids (GSL). *B3GNT5* initiates the synthesis of lactotriosylceramide by transferring N-acetylglucosamine to the C-3 position of galactose within lactose ceramide. Lactose ceramide synthase is an alternative name for *B3GNT5* due to this function. *B3GNT5*-mediated glycolipid synthesis has been reported to play significant biological roles in B cell activation [7], preimplantation development, and nervous system development [8, 9] in multiple studies. Elevated levels of *B3GNT5* appear to be significantly associated with the advancement of breast, lung, and ovarian carcinoma [10–12]. *B3GNT5*-mediated glycosphingolipids are essential for the differentiation of acute myeloid leukemia (AML) cells [13].

In this research, we explored the expression of B3GNT5 and its correlation with cancer prognosis by utilizing data from The Cancer Genome Atlas (TCGA), Genotype-Tissue Expression (GTEx), and Cancer Cell Line Encyclopedia (CCLE) databases obtained through the UCSC XENA database. Furthermore, we examined how B3GNT5 expression correlates with immune cell infiltration score, cell cycle, immune checkpoints, immunosuppressive genes, immune-activating genes, chemokine receptors, chemokines, and drug resistance. Our findings offer novel perspectives on the function of B3GNT5 across various cancers, suggesting a potential mechanism by which B3GNT5 influences the tumor microenvironment (TME), cancer immunotherapy, and cancer stem cell (CSC) stemness.

## Materials and methods

### Data collection

We obtained RNA-seq and clinical information from the TCGA and GTEx databases to analyze 33 types of tumors alongside normal tissues. Data on tumor cell lines were sourced from the CCLE database, while information on DNA copy number and methylation was gathered via the cBioPortal database (https://www.cbioportal.org/).

### Survival prognosis analysis and its relationship with clinical stage

We utilized Kaplan-Meier analysis to assess the overall survival (OS) among patients within the TCGA cohort. In addition, univariate Cox regression analysis was performed to ascertain the prognostic significance of B3GNT5 concerning OS, disease-specific survival (DSS), disease-free interval (DFI), and progression-free interval (PFI) across various cancer cases. Statistical analysis and visualization were performed with the "survival" and "survminer" packages in R software version 4.1.1, Hypothesis testing was conducted through Cox regression, with a P-value below 0.05 considered statistically significant.

## Gene set enrichment and gene set variation analyses

We conducted a Pearson correlation analysis to examine the association between B3GNT5 and all genes using TCGA data. Gene set enrichment analysis (GSEA) was conducted using the "clusterProfiler"R package, employing parameters: nPerm set to 1,000, minGSSize to 10, maxGSSize to 1,000, and a p-value threshold of 0.05. Genes associated with B3GNT5 and a P-value less than 0.05 were selected, and GSEA was carried out using gene sets from the Reactome pathway database. To identify pathways most closely related to B3GNT5, we utilized the 'GSVA' R package to perform gene set variation analysis (GSVA). We categorized the 33 tumor types into two categories based on the median expression levels of B3GNT5 (high versus low expression). The reference genes was obtained from the Molecular Signature Database (MSigDB; http://software.broadinstitute.org/gsea/msigdb/index.jsp), with statistical significance defined as $P < 0.05$."

## Immune cell infiltration

We acquired immune cell infiltration scores related to TCGA data from the TIMER2 and ImmuCellAI databases (http://timer.cistrome.org/ and http://bioinfo.life.hust.edu.cn/web/ImmuCellAI/). To evaluate the extent of immune cell infiltration, patients were categorized into two groups (B3GNT5 high-expression and low-expression) for each TCGA tumor type based on the median *B3GNT5* expression level.

**Association of B3GNT5 with IC50 values of anti-tumor drugs.** Using over 1,000 cancer cell lines, we assessed response data for 192 anti-tumor drugs. The association between *B3GNT5* expression levels and the IC50 of these drugs was illustrated with the R package "ggplot2".

## Cell lines

The Panc-1 cell line, initially obtained from ATCC USA, was provided by Renyi Qin from the Affiliated Tongji Hospital in China. It was cultured in RPMI-1640 medium, enriched with 10% fetal bovine serum (FBS), L-glutamine, and 1% penicillin/streptomycin, and incubated at 37˚C in a 5% CO2 atmosphere.

## Transfections

Panc-1 cells were seeded in six-well plates and transfected with sh-*B3GNT5* and an empty vector (Negative Control). Lentiviruses encoding sh-*B3GNT5* were obtained from Genechem (Shanghai, China) and the transfections were performed based on the manufacturer's guidelines. Lipofectamine was selected as transfection reagent which we purchased from Thermo Fisher Scientific (USA).

## Suspension sphere culture and differentiation

As previously described [14], Lentivirus-transduced PANC-1 cells (1000 cells/mL) were cultured in suspension using serum-free DMED-12 medium (Hyclone, Logan, UT, USA) supplemented with B27 (1:50; Invitrogen, Carlsbad, CA, USA), 20 ng/mL epidermal growth factor (PeproTech EC, London, UK), and 100 ng/mL basic fibroblast growth factor. (PeproTech) [13].

## Statistical analysis

The data were presented as mean ± standard error of the mean and differences between groups were evaluated using a two-tailed Student's t-test. Statistical analysis was conducted with R program version 4.1.1, considering a significance level of $P < 0.05$. (two-tailed).

## Results

### Analysis of B3GNT5 expression variability and correlations across pan-cancer

We analyzed the expression levels of *B3GNT5* among various cancer types using GTEx database data as controls. Our findings demonstrated that upregulation of *B3GNT5* in eighteen different cancers, including cervical squamous cell carcinoma and endocervical adenocarcinoma (CESC), cholangiocarcinoma (CHOL), colon adenocarcinoma (COAD), esophageal carcinoma (ESCA), glioblastoma multiforme (GBM), head and neck squamous cell carcinoma (HNSC), kidney chromophobe (KICH), kidney renal clear cell carcinoma (KIRC), kidney renal papillary cell carcinoma (KIRP), acute myeloid leukemia (LAML), lower-grade glioma (LGG), liver hepatocellular carcinoma (LIHC), lung squamous cell carcinoma (LUSC), pancreatic adenocarcinoma (PAAD), rectum adenocarcinoma (READ), stomach adenocarcinoma (STAD), uterine corpus endometrial carcinoma (UCEC), and uterine carcinosarcoma (UCS). On the other hand, we observed downregulation of *B3GNT5* in five tumors which include breast invasive carcinoma (BRCA), prostate adenocarcinoma (PRAD), skin cutaneous melanoma (SKCM), testicular germ cell tumor (TGCT), and thyroid carcinoma (THCA) (Fig 1A). *B3GNT5* expressional abundance in multiple cancer forms establishes its oncogenic significance, with ESCA, LUSC and HNSC demonstrating the highest expression levels (Fig 1B). The gene expression analysis was conducted on normal human tissues obtained from the GTEx database, and a comparative was conducted to assess the relative expression levels of *B3GNT5*. The findings revealed that this gene was most highly expressed in lung, muscle, and bone marrow tissues (Fig 1C). In various cancer cell lines, the expression of B3GNT5 was evaluated, revealing that the HNSC, ESCA, small cell lung cancer (SCLC), and PAAD cell lines exhibited the highest levels of *B3GNT5* expression (Fig 1D).

We conducted an investigation into the correlation between B3GNT5 expression levels and the clinical significance of cancer therapies. Through our analysis of various stages of cancer defined by the World Health Organization (WHO) and *B3GNT5*, we discovered that the higher the stage in KICH, LIHC, LUAD, PAAD, THCA, and UCEC, the greater the expression of *B3GNT5* (Fig 2A–2F). These findings indicate a notable reduction in *B3GNT5* expression as stages advance in BRCA, COAD, MESO, and SKCM (Fig 2G–2J). Hence, the aberrant *B3GNT5* expression observed in cancer cells may be closely linked to cancer progression and prognosis.

### Comprehensive analysis of B3GNT5 genetic alterations and their correlations in pan-cancer

We conducted a thorough analysis on alterations in pan-cancers, encompassing a total number of 10,953 patients, to identify potential genetic modifications of *B3GNT5* that could be linked to tumorigenesis. Our analysis detected genetic alterations (including missense mutations, amplification, deep deletion, truncating mutations, and structural variants) in around 6.0% of the cases (S1 Fig). The most common changes across various cancer types were amplifications of *B3GNT5* gene, followed by mutations and deep deletions (Fig 3A). In LUSC, ESCA, UCS, CHOL, HNSC, LUAD, CESC, KICH, PAAD, READ, bladder urothelial carcinoma (BLCA), STAD, BRCA, TGCT, SARC, COAD, KIRP, OV, PCPG, LGG, GBM, PRAD, and SKCM, there was a positive correlation was observed between the copy number and *B3GNT5* expression levels. However, in the case of UVM, the relationship between copy number and B3GNT5 expression was negative (Fig 3B). In THCA, LAML, CESC, COAD, BLCA, KICH, MESO, UVM, UCEC, HNSC, ESCA, ACC, KIRP, PCPG, STAD, PAAD, LIHC UCS, SKCM, LUSC, Thymoma (THYM), LGG, and GBM, it was found that the degree of

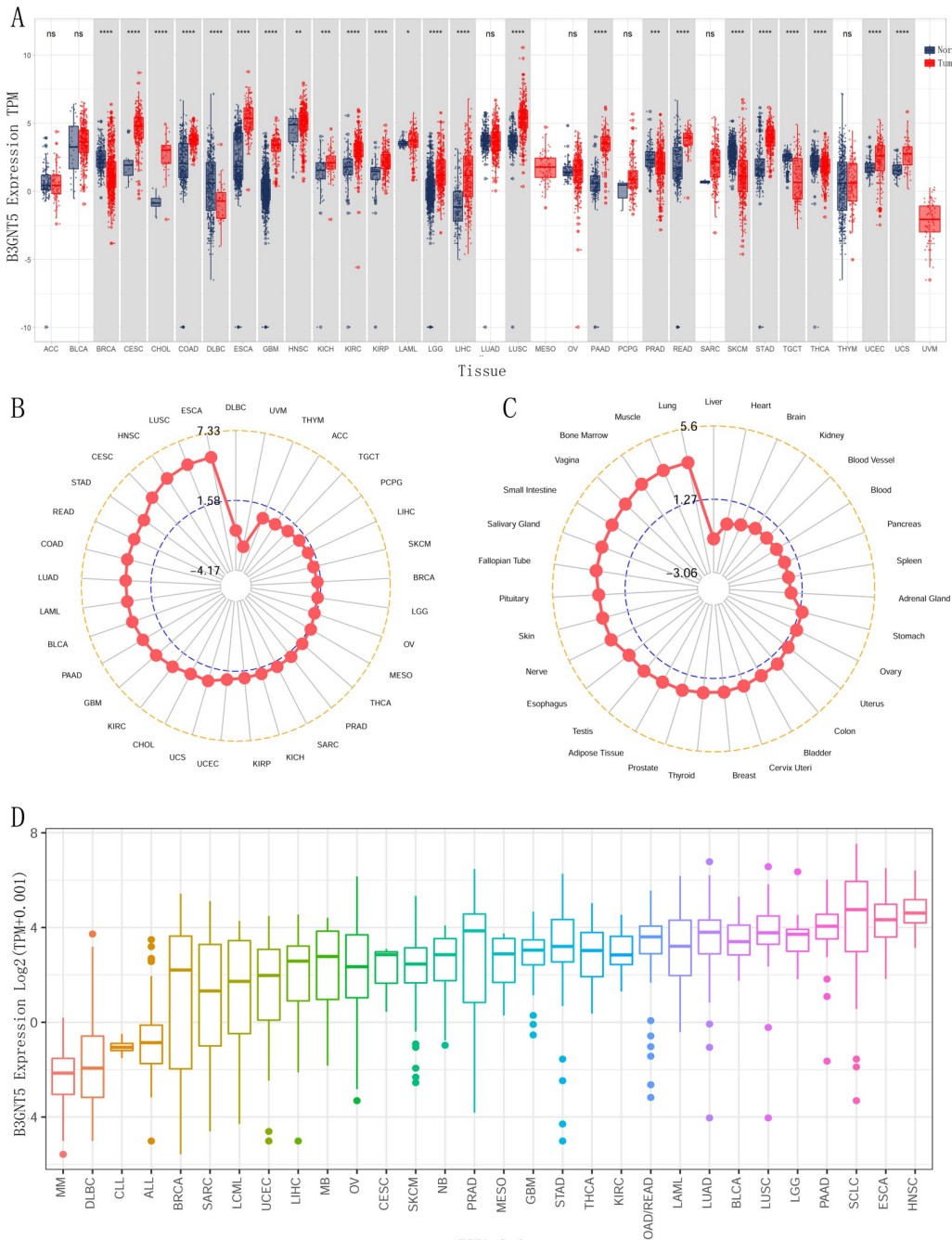

**Fig 1.** The expression levels of *B3GNT5* across pan-cancer were detailed as follows: (A) *B3GNT5* expression in tumor tissues from The Cancer Genome Atlas (TCGA) and normal tissues from TCGA and Genotype-Tissue Expression (GTEx) databases. (B) *B3GNT5* expression in tumor tissues from the TCGA database; (C) *B3GNT5* expression in normal tissues from the GTEx database. (D) *B3GNT5* expression in tumor cell lines from the Cancer Cell Line Encyclopedia (CCLE) database, with mean values represented by datapoints. Statistical significance was denoted by *p < 0.05, **p < 0.01, ***p < 0.001, and ****p < 0.0001, with "ns" indicated not significant.

methylation in the promoter region of *B3GNT5* is adversely linked with the its expression. Conversely, a positive correlation between methylation levels and B3GNT5 expression was identified exclusively in OV (Fig 3C).

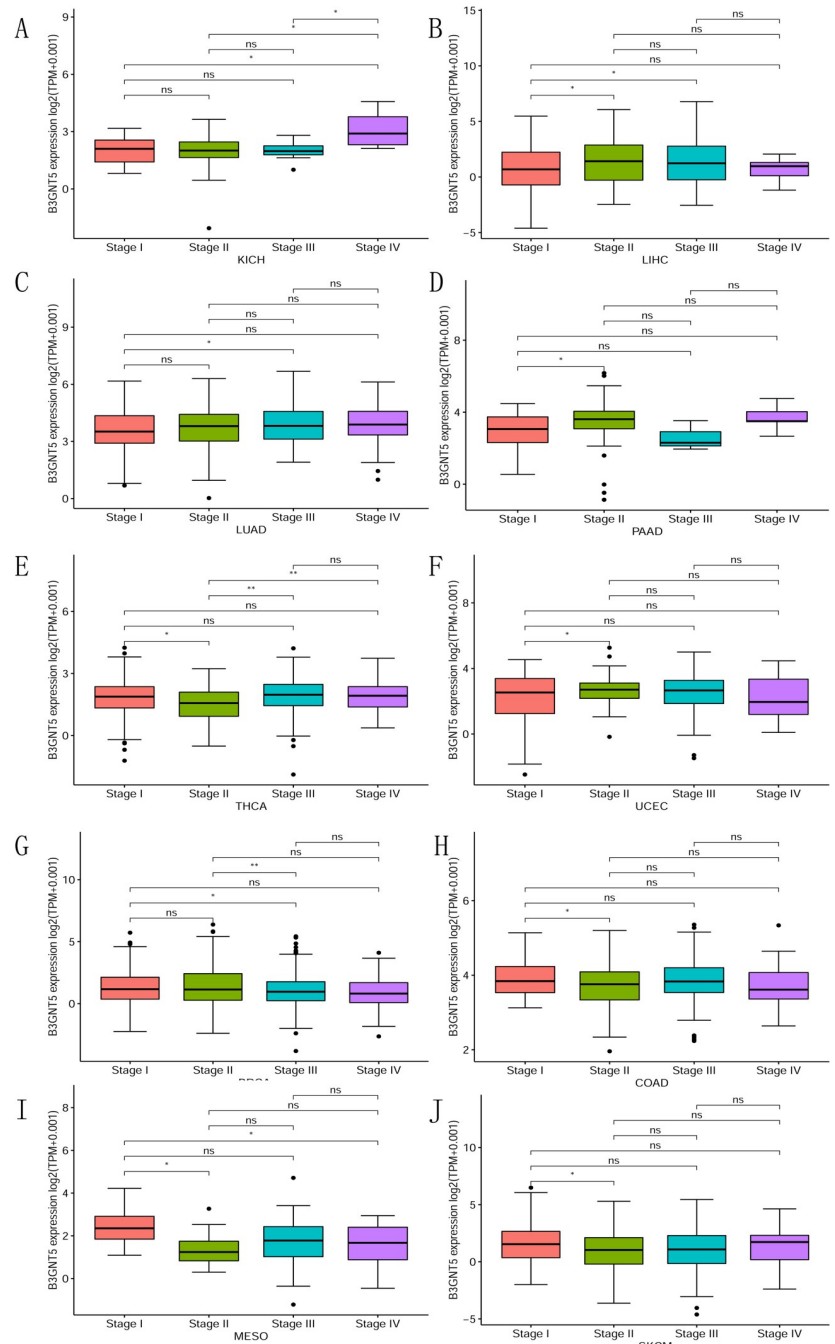

**Fig 2.** For pan-cancer B3GNT5 expression across various World Health Organization (WHO)-defined cancer stages, the differential expression for specified tumor types from the TCGA database is illustrated in Fig A–J. Statistical significance is marked by *p < 0.05, **p < 0.01, ***p < 0.001, ****p < 0.0001, with 'ns' denoting not significant.

## Impact of B3GNT5 expression on prognosis and survival across pan-cancer

We investigated the possible impacts of *B3GNT5* expression on prognosis by scrutinizing its correlation with patient survival. By performing Kaplan-Meier OS analysis, we found that *B3GNT5* was a determinative element for the prognosis of HNSC, KIRP, LGG, LIHC, LUAD, MESO, PAAD, SARC, and UVM patients, signifying its potential involvement in these diseases

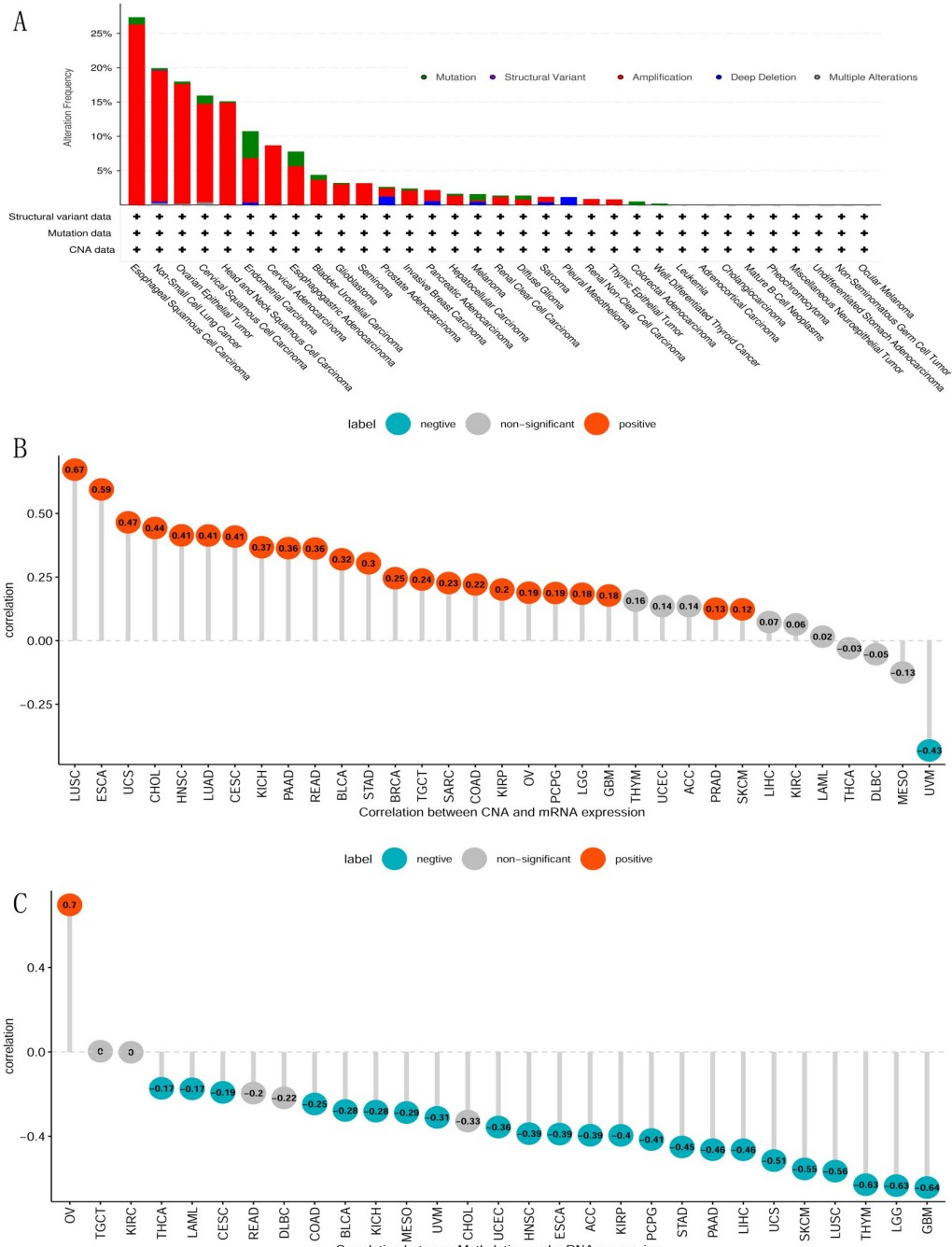

**Fig 3. Analysis of B3GNT5 genetic alterations.** (A) Mutation status across various tumors; (B) the association between *B3GNT5* expression and gene copy number; and (C) the association between *B3GNT5* expression and methylation levels.

(S2 Fig). *B3GNT5* was identified as an element influencing the risk of several cancers, including ACC, HNSC, KICH, KIRP, LGG, LIHC, LUAD, MESO, PAAD, SARC, THCA, THYM, and UVM, as demonstrated by univariate Cox regression analysis (Fig 4A). The DSS analysis determined that *B3GNT5* played a protective role for patients afflicted with PCPG, but acted as a risk element for individuals diagnosed with HNSC, KICH, KIRP, LGG, LIHC, LUAD,

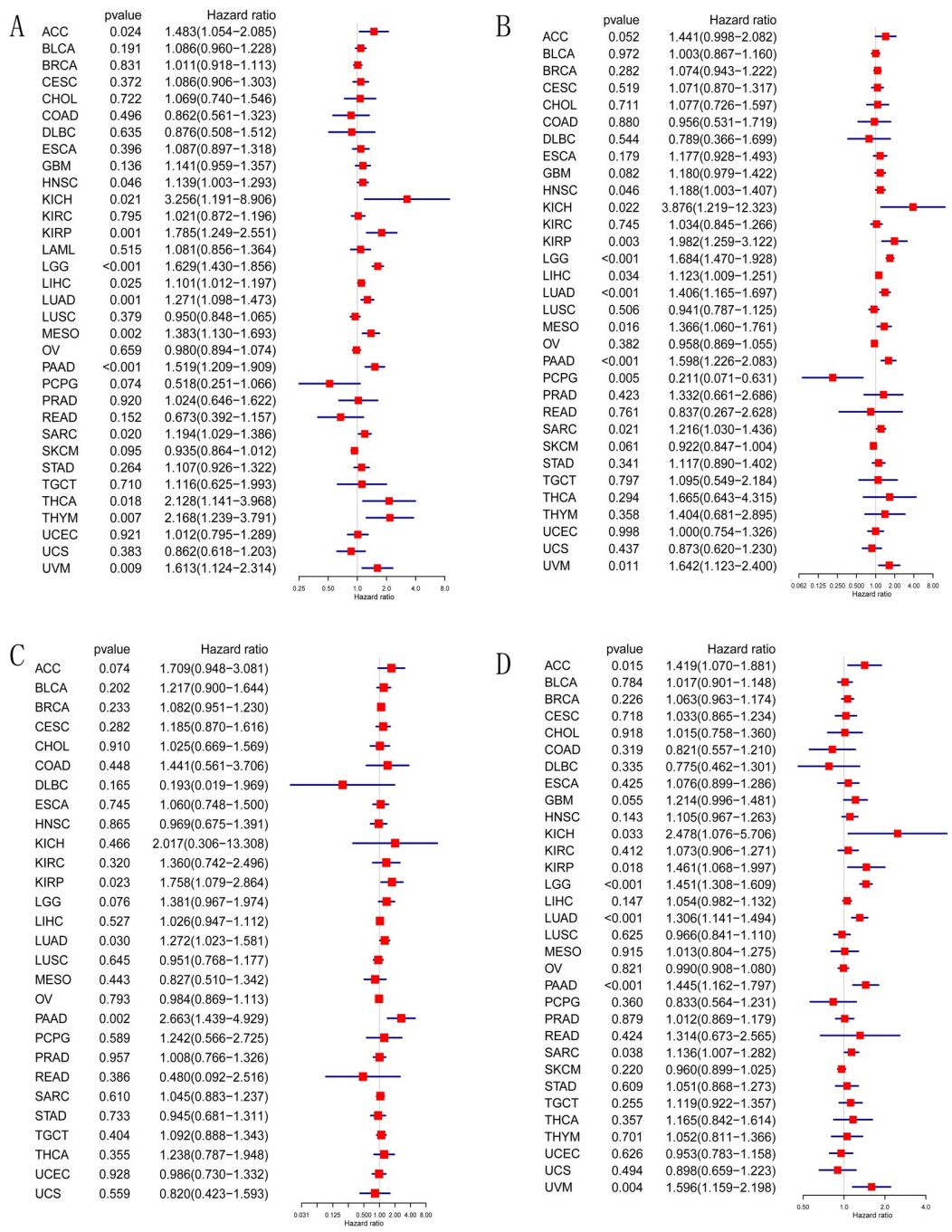

**Fig 4. Univariate Cox regression analysis of *B3GNT5* expression in TCGA pan-cancer.** (A) Forest maps illustrating the association linking *B3GNT5* expression with overall survival (OS), (B) disease-specific survival (DSS), (C) disease-free interval (DFI), (D) and progression-free interval (PFI). Red indicates significant results.

MESO, PAAD, SARC, and UVM (Fig 4B). According to the DFI analysis, the involvement of *B3GNT5* proved risky in cases of KIRP, LUAD, and PAAD (Fig 4C). In accordance with the PFI analysis, *B3GNT5* stood as a perilous element in instances of ACC, KICH, KIRP, LGG, LUAD, PAAD, SARC, and UVM (Fig 4D).

## B3GNT5-related gene pathways: Insights into cell cycle and immune regulation in *pan-cancer*

We performed a screening of *B3GNT5*-related genes and subjected them to subsequent enrichment analyses to elucidate the mechanism underlying cancer carcinogenesis involving *B3GNT5*. Using GSEA in 33 tumor types sourced from TCGA, we identified specific pathways associated with *B3GNT5*. Our findings revealed that *B3GNT5* is significantly linked to pathways regulating the cell cycle and immune response in several malignancies such as CESC, ESCA, KIRP, LUAD, OV, PAAD, and STAD. Additionally, *B3GNT5* was also strongly associated with the pathways involved in immunoregulatory interactions between lymphocytes and non-lymphocytes, cytokine communication, natural immune response, and acquired immune response in LGG, PCPG, KICH, and BRCA tumors. Therefore, our findings revealed that *B3GNT5* is pivotal in modulating the cell cycle and tumor immune microenvironment in malignant tumor cells (Fig 5A–5F). The GSVA investigation findings evince that *B3GNT5* expression has a correlation with the foremost 50 pathways of the Molecular Signatures Database (MsigDB). We observed that DNA repair, Oxidative phosphorylation, MYC targets, Bile acid metabolism, and K-Ras signaling had an adverse correlation with the GSVA score of *B3GNT5* among the 33 cancer classifications (Fig 5G).

## B3GNT5 expression and its impact on tumor microenvironment and immune cell infiltration across pan-cancer

We stratified the TCGA samples derived from 33 distinct types of tumors into two cohorts according to the median expression of *B3GNT5*. We then performed a comparative analysis of the correlated signature score for each tumor across the elevated and reduced expression levels of *B3GNT5* to explore the plausible roles of *B3GNT5* within the tumor microenvironment (TME). Our findings indicated significant associations between *B3GNT5* and various critical pathways, namely nucleotide excision repair, DNA damage response, mismatch repair, DNA replication, base excision repair, epithelial-to-mesenchymal transition (EMT), immune checkpoints, and CD8-T effector (Fig 6A). We utilized the ESTIMATE algorithm to evaluate stromal and immune cell infiltration across the RNA sequencing profiles of 33 cancer types derived from TCGA database. Our findings indicate that *B3GNT5* expression exhibits significant positive correlations with stromal, ESTIMATE, and immune scores; meanwhile, it displays negative associations with tumor purity scores in LGG, KICH, PCPG, ACC, BRCA, GBM, THCA, and PRAD. Conversely, *B3GNT5* expression is negatively related to stromal, ESTIMATE, and immune scores while positively associated with tumor purity score in ESCA, STAD, and LUSC (Fig 6B).

To gain deeper insights into how B3GNT5 expression affects immune cell infiltration, we performed correlation analyses using two independent sources of immune cell infiltration datasets. According to our findings from the TIMER2 database, B3GNT5 expression showed a positive correlation with levels of effective and resting memory CD4+ T cells, neutrophils, and macrophages (Fig 7A), conversely, it shows a negative correlation with B cells, central memory CD4+ T cells, Th1 CD4+ T cells, NK T cells, and regulatory T cells (Tregs) in the TCGA pan-cancer cohort (Fig 7B). Our analysis using the ImmuCellAI database demonstrated that *B3GNT5* expression had an inverse association with CD8+ T cell infiltration levels in THYM, TGCT, LUSC, HNSC, CESC, STAD, SKCM, SARC, and ESCA, while showing a positive correlation in UVM and ACC (Fig 7C). Furthermore, *B3GNT5* expression was positively linked to infiltration levels of Tregs, macrophages, and neutrophils while negatively related to those of B cells and CD8+ T cells. These findings align with the results from the TIMER2 database and suggested that *B3GNT5* may contribute to decreased infiltration of B lymphocytes and CD8+ T

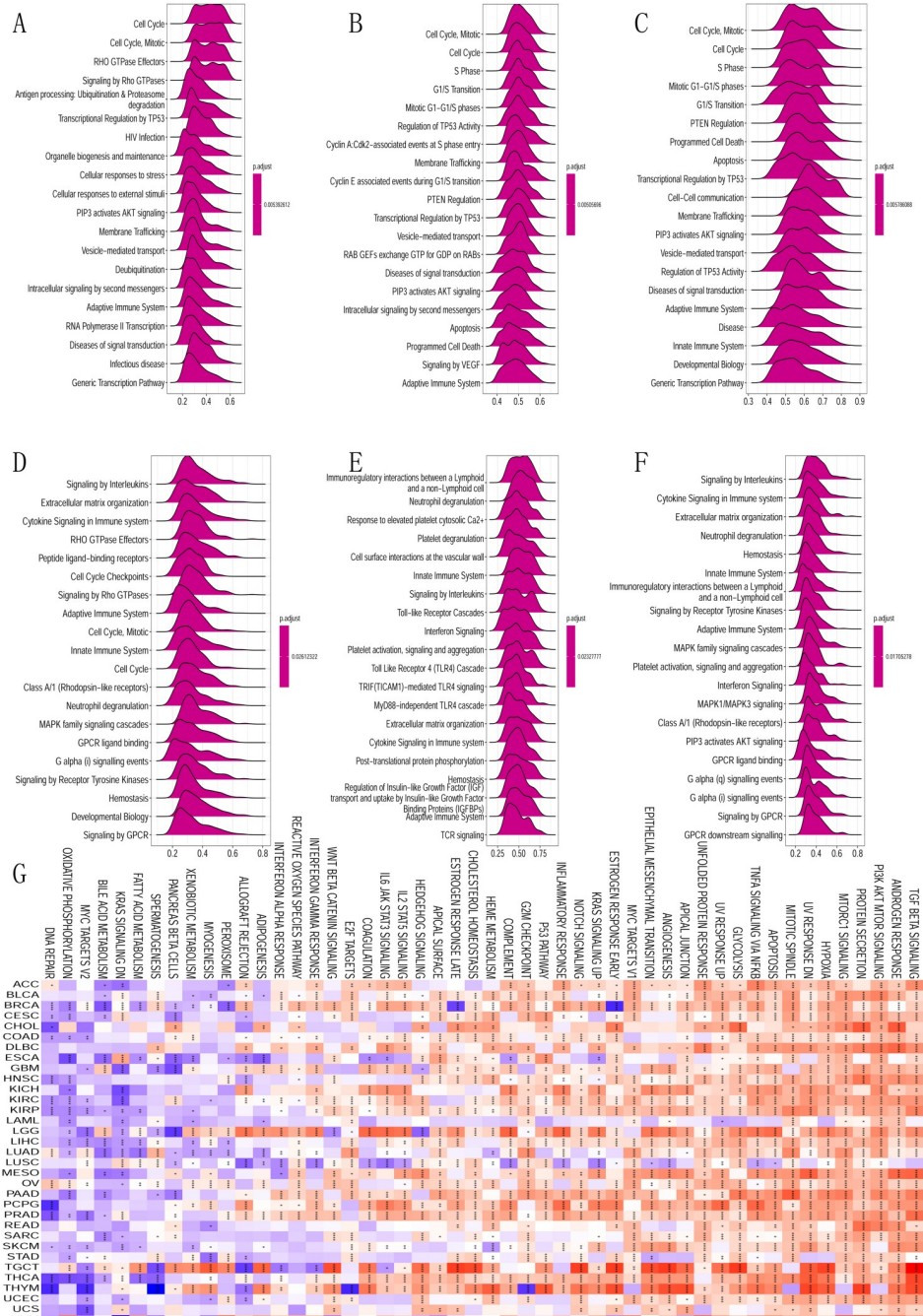

**Fig 5. Gene set enrichment analysis (GSEA) of *B3GNT5* across pan-cancer.** (A–F) Top 20 GSEA terms for the specified tumor types: A: LUAD; B: OV; C: PAAD; D: BRCA; E: LGG; F: PCPG. (G) Gene set variation analysis (GSVA) of *B3GNT5* across pan-cancer with the Top 50 GSEA terms for the indicated tumor types.

lymphocytes, while promoting the accumulation of MDSCs, Tregs, and tumor-associated macrophages (TAMs), thus potentially explaining its role as a risk factor in various tumor types.

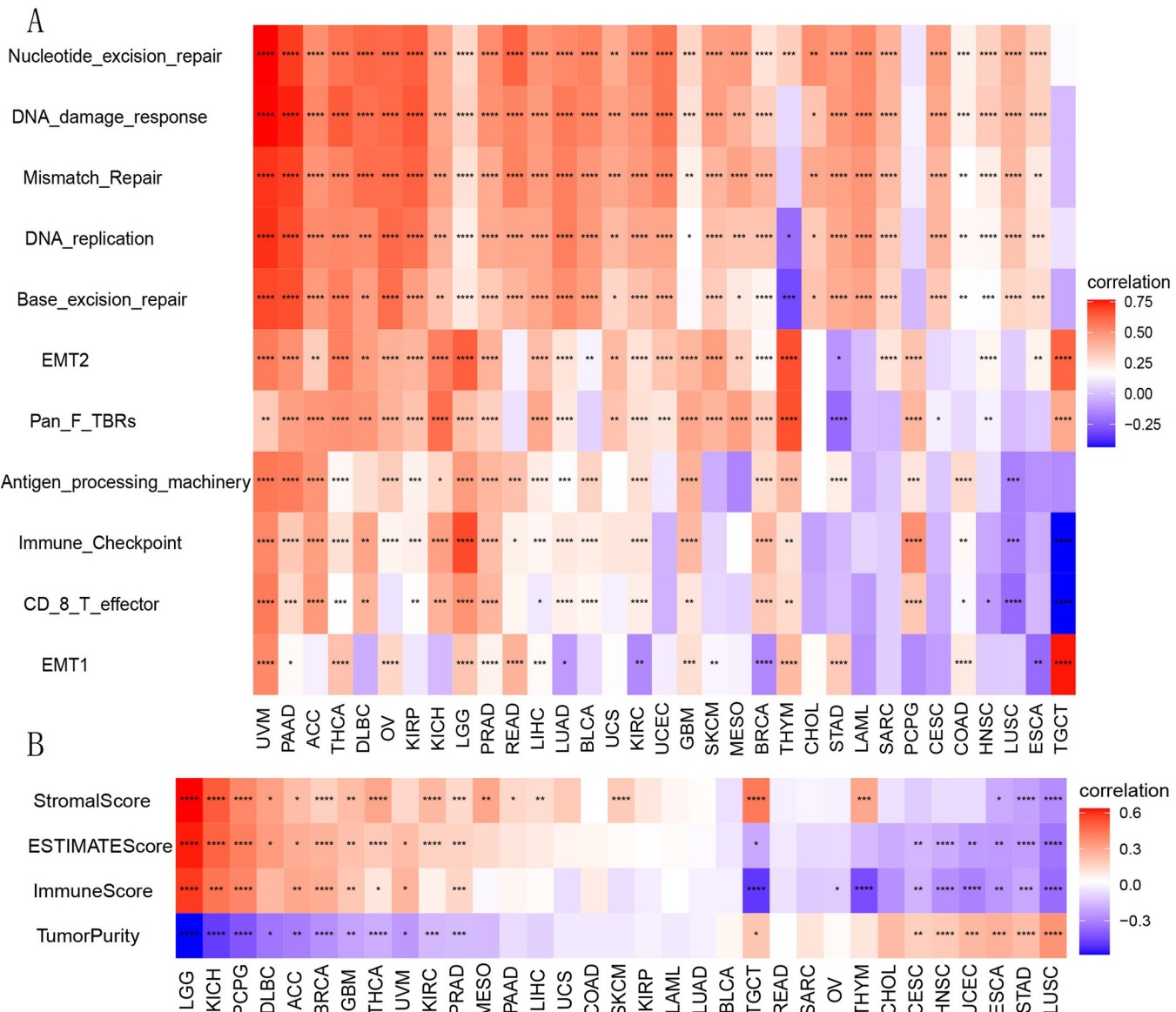

**Fig 6.** (A) The heatmap illustrates the correlation linking *B3GNT5* expression with tumor microenvironment (TME) characterization; (B) ESTIMATE analysis examines *B3GNT5* expression across pan-cancer.

## B3GNT5 expression correlates with immune markers and pathways, highlighting its role across various cancers and immunotherapy

We explored how B3GNT5 levels correlate with a range of immune-related markers, including genes that activate the immune system, those that suppress it, as well as various chemokines and their receptors. Our analysis indicated a positive correlation between *B3GNT5* and immune activation markers, including CD276, PVR, NT5E, STING1, and TNF-SF18 (Fig 9A), as well as immunosuppressive genes TGF-BR1, IL-10PR, KDR, CD274, PDCD1LG2, IL-10, and IDO1 in the pan-cancer cohort (Fig 7D). Furthermore, our results indicated a close association between *B3GNT5* expression and immune checkpoints across various cancer types in the TCGA database (Figs 8 and S3). Moreover, our analysis demonstrated a positive association

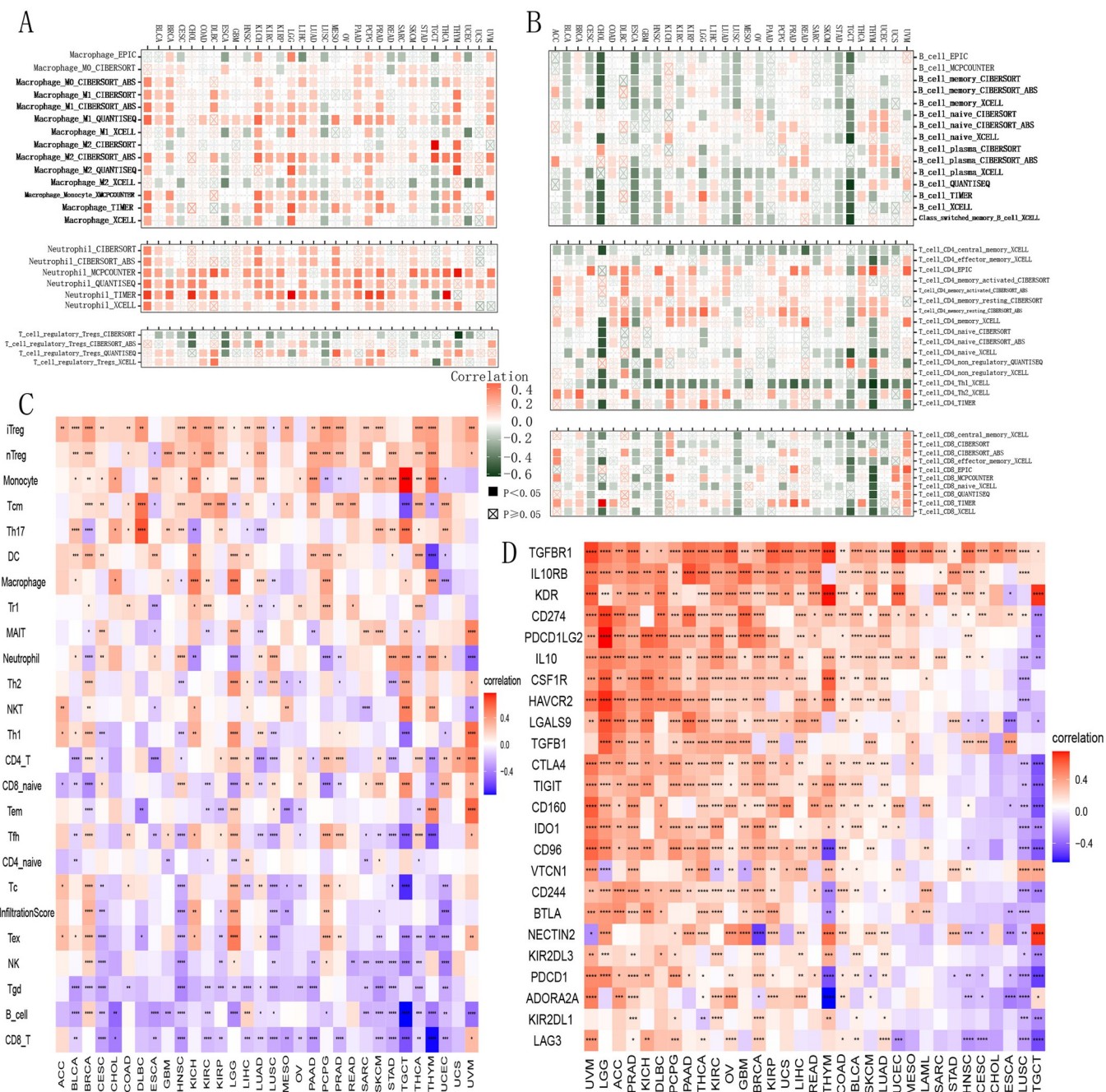

**Fig 7. Immune cell infiltration analysis.** (A) The association linking *B3GNT5* with infiltration levels of macrophage lymphocytes, neutrophil cells, and regulatory T lymphocytes (Tregs) using TIMER2 data. (B) The association linking *B3GNT5* with infiltration levels of B cells, CD4+ T lymphocytes, and CD8 + T lymphocytes using TIMER2 data. (C) Association between *B3GNT5* and infiltration level of the indicated immune cells through ImmuCellAI data. (D) The heatmap illustrates the relationship linking *B3GNT5* expression with immunosuppressive status-related genes. *p < 0.05, **p < 0.01, ***p < 0.001, and ****p < 0.0001.

between *B3GNT5* expression and chemokines including CXCL8, CXCL5, and CXCL16, as well as chemokine receptors such as CXCR2, CCR1, and CCR8 (Fig 9B and 9C). Additionally, we generated heatmaps to visualize the association between *B3GNT5* expression and genes related to pyroptosis (Fig 9D), major histocompatibility complex (MHC) (Fig 9E), autophagy

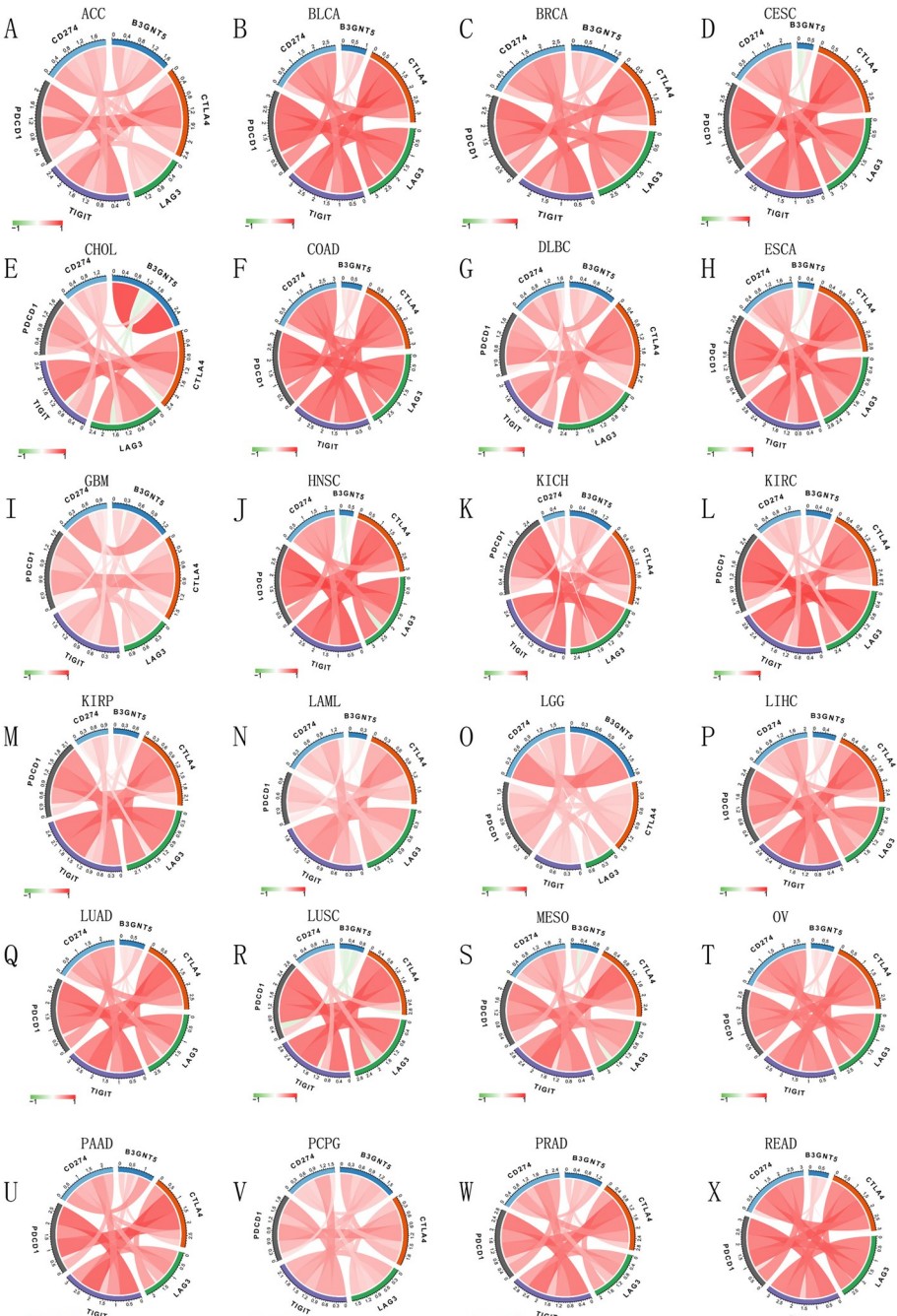

**Fig 8. Immune cell infiltration analysis.** The associationg linking *B3GNT5* expression with the immune checkpoints in ACC, BLCA, BRCA, CESC, CHOL, COAD, Lymphoid Neoplasm Diffuse Large B-cell Lymphoma (DLBC), ESCA, GBM, HNSC, KICH, KIRC, KIRP, LAML, LGG, LIHC, LUAD, LUSC, MESO, OV, PAAD, PCPG, PRAD, and READ (A–X).

(S4A Fig), ferroptosis (S4B Fig), M6A (S4C Fig), EMT upregulation (S5A Fig), EMT downregulation (S5B Fig), TGF-β1 signaling (S6A Fig), and Wnt-β1-catenin signaling (S6B Fig). These results offer significant evidence regarding the potential mechanisms by which B3GNT5 influences cancer progression and immune-based therapies.

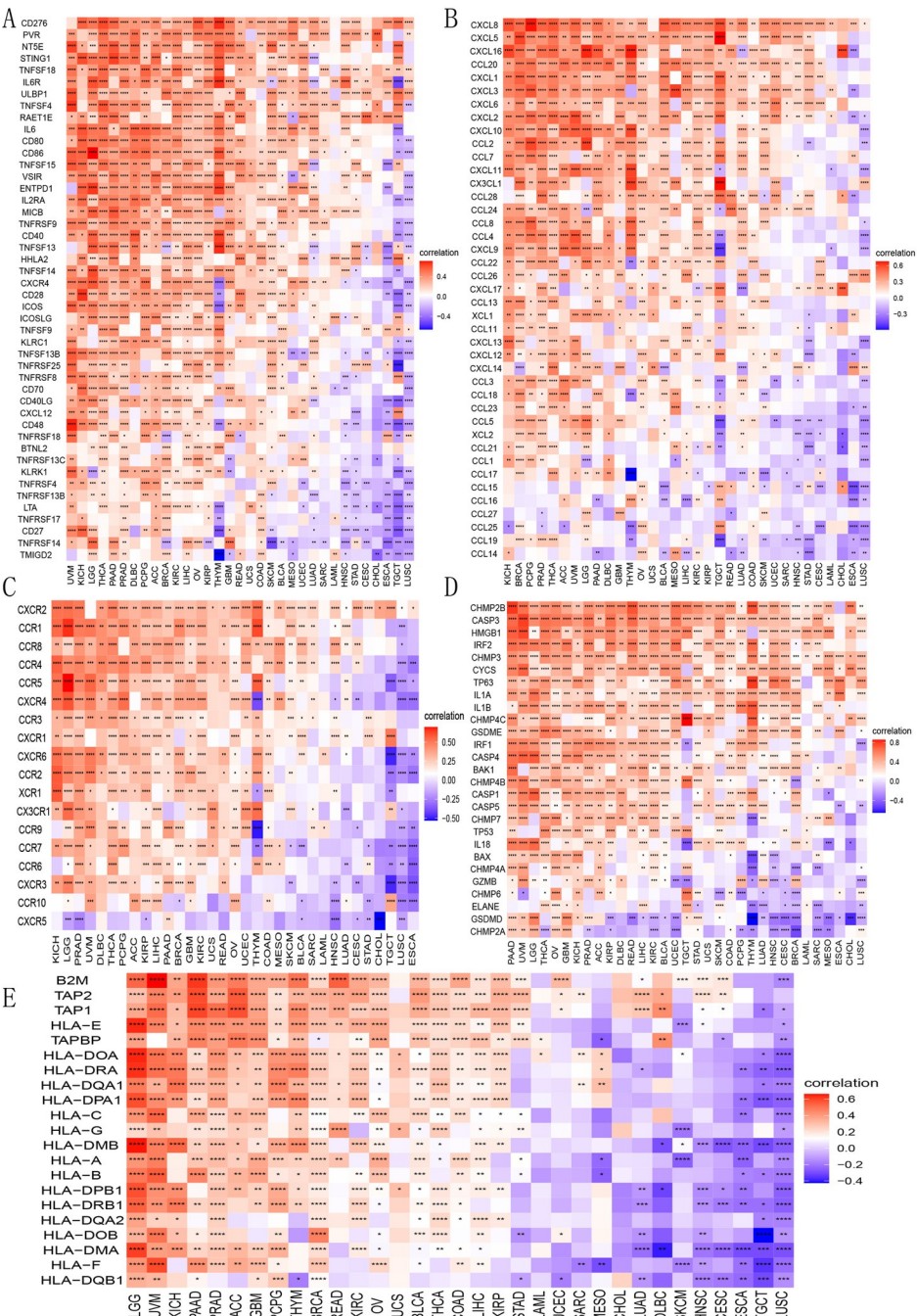

**Fig 9. Heatmaps presenting the association between *B3GNT5* expression and immunoregulation correlated genes.**
(A) Genes of immune activation, (B) chemokine genes, (C) chemokine receptor genes, (D) pyroptosis genes, and (E) major histocompatibility complex (MHC) genes. *p < 0.05, **p < 0.01, ***p < 0.001, and ****p < 0.0001.

## B3GNT5 expression correlates with sensitivity to key anticancer drugs: A comprehensive analysis of IC50 associations

In our study, we evaluated a total of 192 anticancer drugs and identified the IC50 values of 159 drugs showed a significant correlation with *B3GNT5* levels. Based on significant positive or

negative correlations, we selected the top 20 drugs, including Nutlin-3a (r = 0.341, P = $1.915 \times 10^{-22}$), PRIMA-1MET (r = 0.331, P = $5.08 \times 10^{-20}$), Elephantin (r = 0.325, P = $2.35 \times 10^{-19}$), Sabutoclax (r = 0.314, P = $5.46 \times 10^{-18}$), PCI-34051 (r = 0.307, P = $2.32 \times 10^{-17}$), Nilotinib (r = 0.295, P = $1.26 \times 10^{-16}$), AMG-319 (r = 0.287, P = $1.34 \times 10^{-15}$), MIRA-1 (r = 0.2846, P = $2.08 \times 10^{-15}$), Oxaliplatin (r = 0.1948, P = $1.66 \times 10^{-14}$), Fulvestrant (r = 0.1969, P = $1.79 \times 10^{-14}$), PD173074 (r = 0.2739, P = $1.84 \times 10^{-14}$), AZD4547 (r = 0.2714, P = $3.50 \times 10^{-14}$), Sorafenib (r = 0.2704, P = $4.18 \times 10^{-14}$), EPZ004777 (r = 0.2614, P = $6.19 \times 10^{-14}$), I-BRD9 (r = 0.2692, P = $6.65 \times 10^{-14}$), BIBR-1532 (r = 0.2689, P = $7.94 \times 10^{-14}$), Vorinostat (r = 0.2656, P = $1.16 \times 10^{-13}$), MIM1 (r = 0.2607, P = $4.19 \times 10^{-13}$), MK-8776 (r = 0.2526, P = $2.52 \times 10^{-12}$), and Zoledronate (r = 0.2512, P = $6.51 \times 10^{-12}$, Fig 10A–10U).

## Impact of B3GNT5 downregulation on self-renewal in pancreatic cancer cells: A sphere formation study

In order to research the effects of *B3GNT5* heterotopic expression on PAAD cells, we cultured a stable pancreatic cancer cell line (PANC-1) with sh-RNA inhibition of *B3GNT5*. To assess self-renewal ability of cells generated in serum-free conditions, the number and size of spheres were evaluated. The results showed that PANC-1 cells with downregulated *B3GNT5* expression displayed smaller spheres than those in the negative control (NC) group, as shown in Fig 11A. Additionally, the number of cells per sphere exhibited a marked reduction in the Sh-*B3GNT5* group relative to the NC group (Fig 11B, P<0.01). Cells with downregulated *B3GNT5* expression had fewer spheres formed over three passages than those in the NC group (Fig 11C, P<0.01).

## Discussion

Over the past few years, the use of inhibitors targeting immune checkpoints in immunotherapy has become a vital treatment strategy for multiple cancer types, leading to significant advances in cancer therapy [15]. The discovery and development of specific inhibitors for immune checkpoints, including CTLA-4 and programmed cell death protein 1 (PD-1), have revolutionized cancer immunotherapy. However, the efficacy of these therapies has varied significantly among different types of cancers and individuals. While certain malignancies such as lung cancer, breast cancer, and melanoma have exhibited promising outcomes with immunotherapeutic strategies, the potential of immunologically "cold" tumors like PAAD remains unclear. Therefore, it is important to investigate the distinct characteristics and fundamental mechanisms underlying the heterogeneous outcomes of immunotherapy among different cancer types. Cancer stem cells (CSCs) possess unique properties that enable them to evade immune recognition and elimination [16]. Multiple studies have recently established that CSCs are capable of shaping the immunosuppressive and tumor-promoting environment of TME via modulation of various immune cells, contributing to resistance towards immunotherapeutic approaches. Identifying the critical binding target between CSCs and cancer immunotherapy would be a significant advancement in this field. Given the potential for enhanced tumor cell immunogenicity and T cell activation, inhibition of GSL synthesis through suppression of *B3GNT5* expression may be implemented as a complementary approach along with current immunotherapeutic strategies, including PD-1 blockade [17]. Study has indicated a noteworthy decrease in *B3GNT5* expression during differentiation of glioblastoma stem cells [18], suggesting that *B3GNT5* could potentially serve as a connecting link.

In this research, we performed an initial assessment of *B3GNT5* expression and its prognostic relevance across multiple cancer types, revealing high expression levels to be present in 18

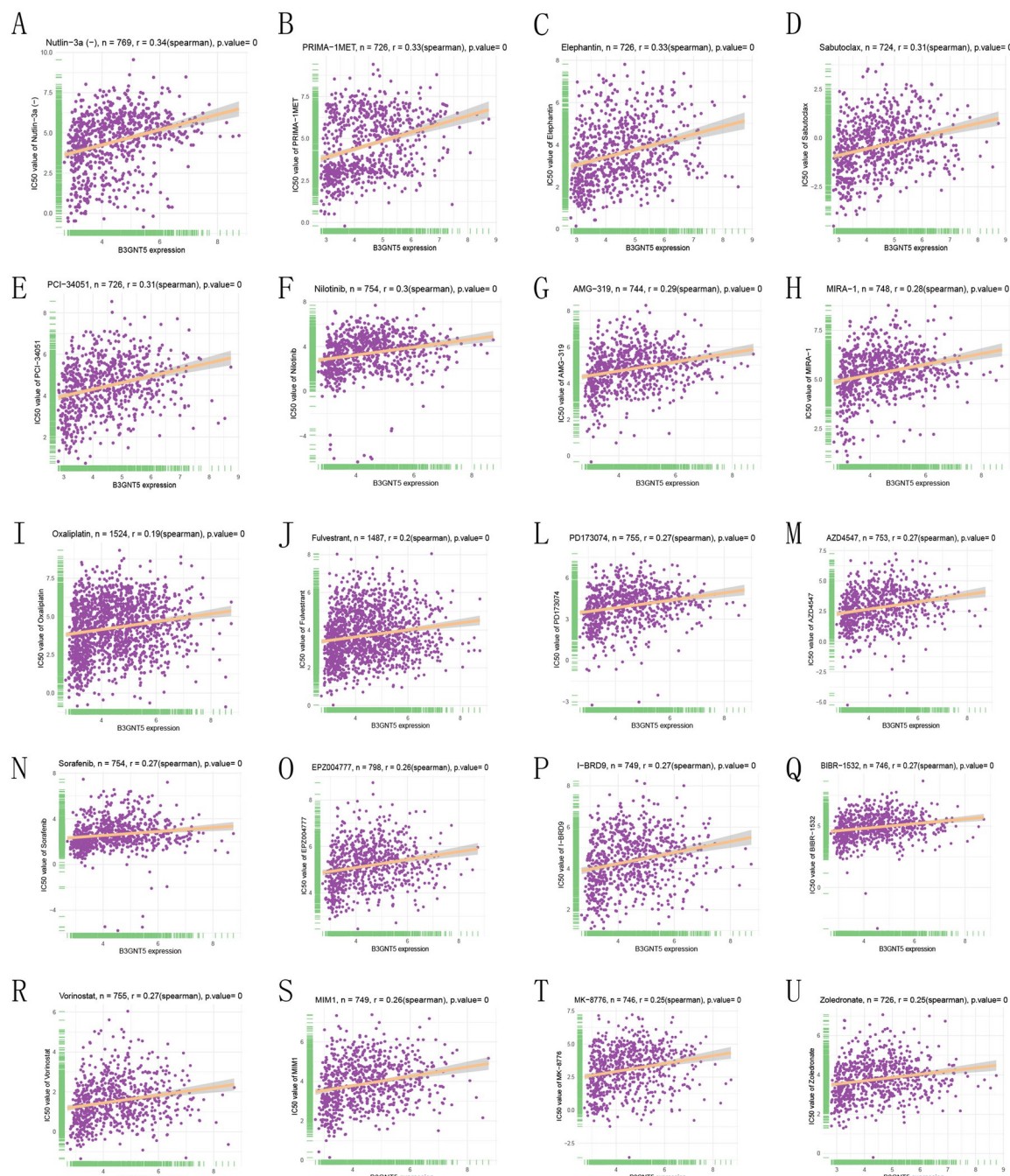

**Fig 10. Drug resistance analysis.** The association between *B3GNT5* expression and IC50 for different anticancer drugs (A–U).

of these tumors. Upon conducting Kaplan-Meier overall survival analysis, we identified *B3GNT5* as a risk element for patients across nine tumor types. Furthermore, our univariate Cox regression analysis disclosed that this gene acted as a risk element in twelve different tumor types. Similarly, *B3GNT5* was implicated as a protective element in patients with PCPG but as a risk factor across eleven other tumor types according to our DSS analysis. These consistent findings indicate that *B3GNT5* may possess proto-oncogenic properties across a

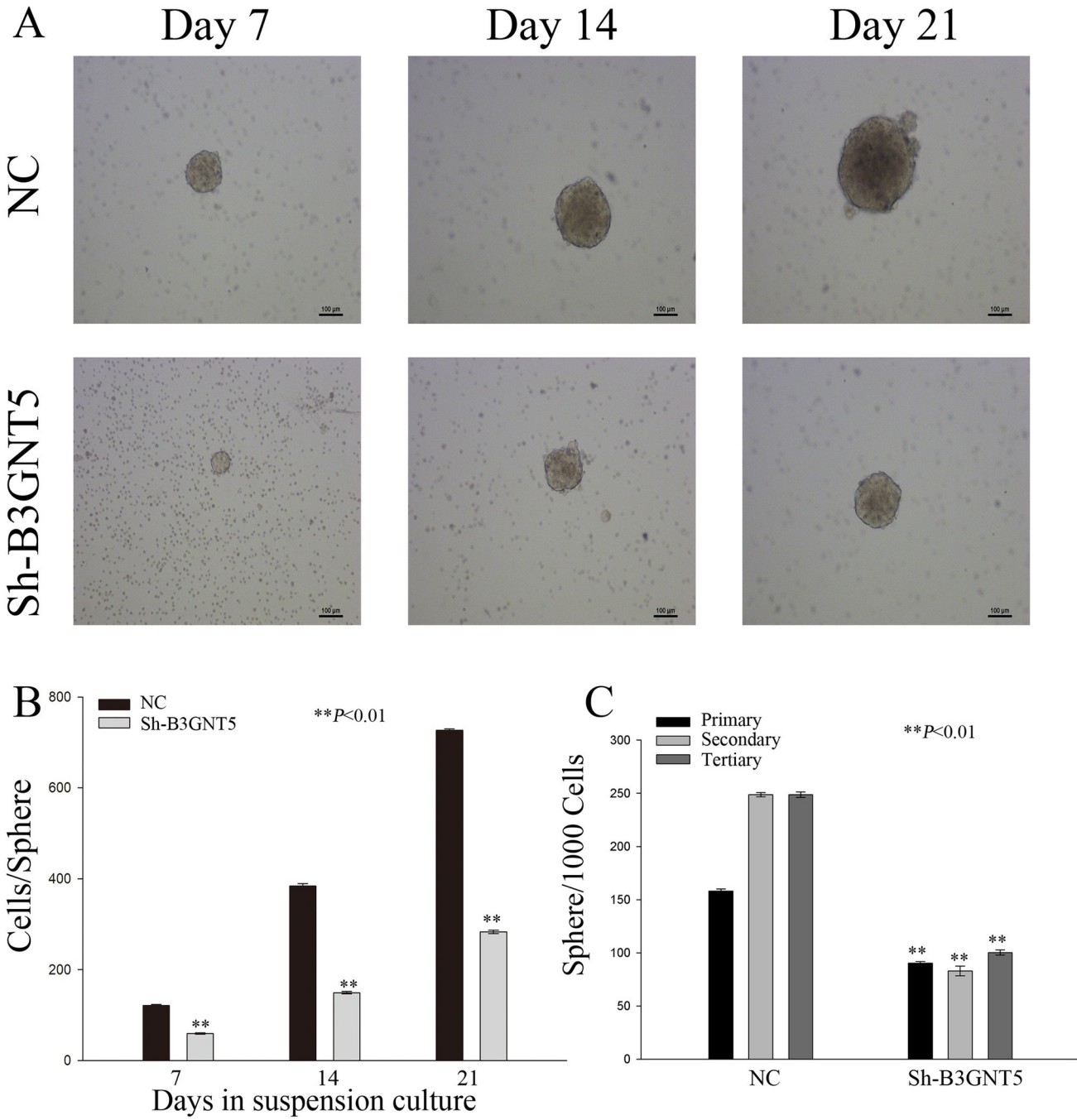

**Fig 11. Modulating B3GNT5 expression influences the self-renewal capability of PAAD cells in vitro.** (A) Sphere formation was observed in Panc-1 cells with B3GNT5 suppression at various time intervals. (B) Quantification of cell numbers within each sphere was conducted. (C) The number of spheres formed per 1000 cells reflects sphere-forming ability, relative to negative control group. **P < 0.01. NC denotes the negative control group; sh-B3GNT5 refers to small hairpin B3GNT5.

majority of tumor types. By means of gene set enrichment analysis (GSEA) involving *B3GNT5*, we successfully identified strong correlations between this gene and numerous pathways related to cell cycle control and immune signaling, particularly immunoregulatory interactions between lymphocytes and non-lymphocytes, which encompasses a total of 135 genes, inclusive

of receptors and cellular adhesion molecules that regulate the response of lymphocyte-related cells (i.e., B-, T-, and NK lymphocytes) towards to self-antigens, tumor antigens, and also to pathogens [19–23]. The findings of this investigation suggest a strong link between *B3GNT5* and the regulation of immune microenvironment in tumors, as well as ligand-receptor communication between malignant cells and lymphoid cells. Prior research has demonstrated that such immune cells can maintain the stemness and viability of cancer stem cells (CSCs) [24]. In our study, we observed that decreased *B3GNT5* expression negatively impacts the self-renewal capacity of PAAD cells. Consequently, these results infer that the influence of *B3GNT5* on immune cells within the TME plays a regulatory role in CSC stemness and malignant features. Further research is needed to explicate the underlying mechanisms. CD8+ T cells, which belong to the T lymphocyte population, are cytotoxic killer cells essential for cell-mediated immunity, particularly within tumor tissues [25, 26]. The activation and formation of memory in cytotoxic CD8+ T lymphocytes are dependent on CD4+ T lymphocytes [27]. Multiple studies have found strong correlations between CSC stemness and CD8+ T cells in multiple cancer types [28]. For instance, CSCs can produce TGF-β and CCL2, which inhibit CD8+ and CD4+ T cell activation and proliferation [29]. Furthermore, certain chemokine family members such as CCL1, CCL2, and CCL5, which are highly expressed by CSCs in different cancer types, stimulate the infiltration of T-reg cells into the tumor microenvironment [29–31]. Moreover, in prostate cancer, CSCs secrete tenascin-C to hinder the activation and proliferation of CD8+ and CD4+ T cells through interaction with α5β1 integrin located on T cells [32]. Interestingly, T cells also regulate CSC stemness. Low IFN-γ levels activate the PI3K/AKT/NOTCH1 pathway and promote CSC stemness. In NSCLC, CD8+ T cells mainly produce IFN-γ [33]. Our analysis using two distinct data sources indicates that *B3GNT5* expression is inversely associated with CD8+ T lymphocytes, CD4+ T lymphocytes, and natural killer cells, which could help shed light on the connection between *B3GNT5* and diverse tumor types. T-reg cells prevent dangerous tumor cells from attack by cytotoxic CD8+ T lymphocytes [34, 35]. Many research pieces demonstrate that T-reg cells secrete IL-10, thereby promoting leukemic stem cell stemness through activation of PI3K/AKT/OCT4/NANOG pathways in AML [36]. Additionally, our immune cell infiltration data suggest that Treg infiltration levels and *B3GNT5* expression are positively correlated, implying that even a large group of cytotoxic CD8+ T cells' functioning is limited, and B cells are responsible for antigen presentation. Activated B cells from a control donor's peripheral-blood lymphocytes (PBL) present antigens to CD4+ and CD8+ T cells [37, 38]. This selective presentation of cognate antigens using surface Ig molecules leads to tumor-infiltrating B cells (TIL-Bs) delivering antigens more efficiently than tumor dendritic cells (DCs). DCs excite CD4+ and CD8+ TILs in the lymph nodes, followed by TIL-Bs initiating recall responses in the tumor. TIL-Bs act as local antigen-presenting cells (APC) that provide secondary stimulation to CD4+ TILs, allowing their survival and proliferation for an extended period [39, 40]. Interestingly, *B3GNT5* expression negatively correlates with B cell and CD8+ T cell infiltration levels while positively correlating with immune-activating and immunosuppressive genes across pan-cancer, which supports the potential role of *B3GNT5* as an immune checkpoint molecule and a focal point of the CSC-immune cell crosstalk. We surmise that *B3GNT5* regulates signal pathways or cell death processes, such as EMT, Wnt-β1-catenin, TGF-β1 signaling, pyroptosis, autophagy, or ferroptosis, which have all been related to *B3GNT5* in our analysis, to impact CSC-immune cell crosstalk and carcinogenic biological properties.

Our study has several limitations that must be acknowledged. Additional experiments are necessary to evaluate the mechanisms by which *B3GNT5*-mediated interactions of cancer stem cells (CSCs) with immune cells, as well as to validate the potential of *B3GNT5* as an immune checkpoint target in clinical trials. To conclude, we conducted a comprehensive analysis of *B3GNT5* across various cancer types and highlighted its potential significance in regulating the immune response and serving as a prognostic indicator for patients. B3GNT5 is capable of suppressing the activation and proliferation of T lymphocytes by promoting the secretion of TGF-β and CCL2 by CSCs (cancer stem cells). These factors not only inhibit the function of effector T cells, but also enhance the infiltration of regulatory T cells (Tregs) into TME, thus inhibiting the anti-tumor immune response. B3GNT5 may become one of the risk factors of most tumors by promoting the accumulation of cells that suppress the immune response, such as Tregs and tumor-associated macrophages (TAMs). Abnormal synthesis of GSLs mediated by B3GNT5 may alter the glycosylation pattern on tumor cell membranes, subsequently impacting immune cell recognition and function.For example, in pancreatic cancer cell line PANC-1, a decrease in cell self-renewal ability was observed after down-regulation of B3GNT5 expression, indicating the importance of B3GNT5 in maintaining the characteristics of tumor stem cells. In addition, B3GNT5 is also associated with key pathways such as DNA repair pathway, epithelial-mesenchymal transition (EMT) and immune checkpoints. Abnormal activation or inhibition of these pathways may further aggravate the phenomenon of tumor immune escape, so that tumor cells can better adapt to unfavorable environment and escape the attack of host immune system. *To sum up, B3GNT5 affects tumor immune response through many mechanisms, and its role in tumor occurrence and development can not be ignored. In the future, more experimental data are needed to further explore the specific mechanism of B3GNT5 and its possibility as a potential therapeutic target.*

## Supporting information

**S1 Fig. Genetic alteration analysis of *B3GNT5*: The mutation status of *B3GNT5* in different tumors.**
(TIF)

**S2 Fig. Link between Kaplan–Meier overall survival estimates and B3GNT5 expression levels.** (A–I) The connection between B3GNT5 expression and Kaplan–Meier overall survival across Pan-Cancer types from the TCGA database is shown. The median B3GNT5 expression value for each tumor type was used as the threshold value.
(TIF)

**S3 Fig. Immune cell infiltration analysis.** The correlation linking *B3GNT5* expression with immune checkpoints in SARC, SKCM, STAD, TGCT, THCA, THYM, UCEC, UCS, and UVM (A–I).
(TIF)

**S4 Fig. Heatmaps presenting the association linking *B3GNT5* expression with immunoregulation-related genes.** (A) Autophagy genes, (B) ferroptosis, and (C) M6A. $^*$p $< 0.05$, $^{**}$p $< 0.01$, $^{***}$p $< 0.001$, and $^{****}$p $< 0.0001$.
(TIF)

**S5 Fig. Heatmaps presenting the association between *B3GNT5* expression and immunoregulation-related genes.** (A) EMT upregulated genes and (B) EMT downregulated genes. $^*$p $< 0.05$, $^{**}$p $< 0.01$, $^{***}$p $< 0.001$, and $^{****}$p $< 0.0001$.
(TIF)

**S6 Fig. Heatmaps presenting the association between *B3GNT5* expression and immunoregulation-related genes.** (A) TGF-β1-signaling genes and (B) Wnt-β1-catenin-signaling genes. *p < 0.05, **p < 0.01, ***p < 0.001, and ****p < 0.0001.
(TIF)

## Acknowledgments

The authors express their gratitude to the TCGA and GEO projects for providing access to their data.

## Author Contributions

**Conceptualization:** Feng Peng, Yu Xie, Renyi Qin.

**Data curation:** Feng Peng, Yechen Feng, Shuo Yu, Hebin Wang, Yu Xie.

**Formal analysis:** Shuo Yu, Yu Xie.

**Funding acquisition:** Renyi Qin.

**Investigation:** Feng Peng, Shuo Yu.

**Methodology:** Feng Peng.

**Project administration:** Feng Peng, Ruizhi He, Renyi Qin.

**Resources:** Yechen Feng, Ruizhi He.

**Software:** Ruizhi He.

**Supervision:** Ruizhi He, Hebin Wang.

**Validation:** Feng Peng, Hebin Wang, Yu Xie, Renyi Qin.

**Visualization:** Renyi Qin.

**Writing – original draft:** Feng Peng, Yu Xie.

**Writing – review & editing:** Feng Peng, Yu Xie, Renyi Qin.

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
