## [Decision Letter · Decision Letter 0]

11 Sep 2024

PONE-D-24-23687Pan-cancer Analysis of B3GNT5 with Potential Implications for Cancer Immunotherapy and Cancer Stem Cell stemnessPLOS ONE

Dear Dr. Qin,

Thank you for submitting your manuscript to PLOS ONE. After careful consideration, we feel that it has merit but does not fully meet PLOS ONE’s publication criteria as it currently stands. Therefore, we invite you to submit a revised version of the manuscript that addresses the points raised during the review process.

**ACADEMIC EDITOR:** Thank you for submitting your manuscript to the Journal and as voucan see that the reviewer thinkyour manuscript is interesting and provide valuable comments for youlreference. Please submit the revised manuscript ASAP and also include a rebuttal that would clearly list all the responses to the reviewer's comments.

We look forward to receiving your revised manuscript.

Kind regards,

Zhiwen Luo

Academic Editor

PLOS ONE

Journal Requirements:

1. When submitting your revision, we need you to address these additional requirements. Please ensure that your manuscript meets PLOS ONE's style requirements, including those for file naming. The PLOS ONE style templates can be found at https://journals.plos.org/plosone/s/file?id=wjVg/PLOSOne_formatting_sample_main_body.pdf and https://journals.plos.org/plosone/s/file?id=ba62/PLOSOne_formatting_sample_title_authors_affiliations.pdf 2. We noticed you have some minor occurrence of overlapping text with the following previous publication(s), which needs to be addressed: https://doi.org/10.3389/fimmu.2021.688215 In your revision ensure you cite all your sources (including your own works), and quote or rephrase any duplicated text outside the methods section. Further consideration is dependent on these concerns being addressed. 3. We note that the grant information you provided in the ‘Funding Information’ and ‘Financial Disclosure’ sections do not match.  When you resubmit, please ensure that you provide the correct grant numbers for the awards you received for your study in the ‘Funding Information’ section. 4. PLOS requires an ORCID iD for the corresponding author in Editorial Manager on papers submitted after December 6th, 2016. Please ensure that you have an ORCID iD and that it is validated in Editorial Manager. To do this, go to ‘Update my Information’ (in the upper left-hand corner of the main menu), and click on the Fetch/Validate link next to the ORCID field. This will take you to the ORCID site and allow you to create a new iD or authenticate a pre-existing iD in Editorial Manager.

Additional Editor Comments:

Thank you for submitting your manuscript to the Journal and as voucan see that the reviewer think your manuscript is interesting and provide valuable comments for your reference. Please submit the revised manuscript ASAP and also include a rebuttal that would clearly list all the responses to the reviewer's comments.

Reviewers' comments:

Reviewer's Responses to Questions

**Comments to the Author**

1. Is the manuscript technically sound, and do the data support the conclusions?

Reviewer #1: Yes

Reviewer #2: Yes

Reviewer #3: Partly

2. Has the statistical analysis been performed appropriately and rigorously? 

Reviewer #1: Yes

Reviewer #2: Yes

Reviewer #3: Yes

3. Have the authors made all data underlying the findings in their manuscript fully available?

Reviewer #1: Yes

Reviewer #2: Yes

Reviewer #3: Yes

4. Is the manuscript presented in an intelligible fashion and written in standard English?

Reviewer #1: Yes

Reviewer #2: Yes

Reviewer #3: No

5. Review Comments to the Author

Reviewer #1: 1. The study utilized data from multiple databases, yet it is still necessary to clarify whether the sample selection is representative.

2. When conducting gene set enrichment analysis and gene set variation analysis, multiple comparisons might be involved, but the text does not mention whether multiple comparison corrections were applied.

3. In the part "Regulating the expression of B3GNT5 can affect self-renewal of PAAD cells", "NC" is mentioned, but the specific setup of the NC group and the type of control are not detailed. In addition, it is not specified whether the experiment was repeated, the number of repetitions, or whether a control group was set up.

Reviewer #2: The manuscript presents a comprehensive analysis of B3GNT5 across various cancer types, exploring its potential implications for cancer immunotherapy and stemness. The study is timely and relevant. The authors have utilized robust datasets from TCGA and GEO, which strengthens the validity of their findings.

1. Please ensure that page and line numbers are provided in the correct format.

2. Please note that the current lack of clarity of the figures makes them more difficult to read.The figures have to be prepared at high resolution in a clear and meaningful manner. Additionally, providing a more detailed legend would help readers understand the data presented.

3. The font formatting of the references needs to be adjusted. Font style and size should be uniform throughout the manuscript, including references.

4. The introduction provides a good background on B3GNT5; however, it could benefit from a more detailed discussion on the specific mechanisms by which B3GNT5 influences immune responses in tumors.

5. It would be helpful to include more details on the statistical analyses performed, including any software used and specific tests applied.

6. The discussion effectively contextualizes the findings within the broader field of cancer research. Are there specific experimental approaches or clinical trials that could be proposed to confirm the role of B3GNT5 in cancer immunotherapy? This would strengthen the manuscript by providing a pathway for future research.cancer stem cell stemness.

7. The authors need to improve their language and grammar to enhance the flow of the text. For example, the phrase "B3GNT5's expression was highly correlated with different immunoregulatory factors" could be rephrased to "The expression of B3GNT5 shows a high correlation with a range of immunoregulatory factors." This change would enhance the clarity of the statement. In addition, the whole manuscript needs to be checked by native English speakers.

Reviewer #3: - The article should undergo extensive English proofreading.

- Too many full stops. They need to be removed. For example: This phenomenon is particularly visible in underdeveloped countries, where about 82% of the world's population lives.(1).

- In the methodology section: transfection is not sufficiently described... please provide details.

- In the results section, each paragraph should include a sentence summarizing what the results actually mean.

- There are no results confirming that sh-RNA transfection was successful. Please analyze B3GNT5 expression after shRNA silencing. Without confirmation of inhibition, it is not possible to show these results and draw any conclusions.

- The resolution of the figures is too small and the results are blurry.

6. PLOS authors have the option to publish the peer review history of their article (what does this mean?). If published, this will include your full peer review and any attached files.

Reviewer #1: No

Reviewer #2: No

Reviewer #3: No

---

## [Author Response · Author response to Decision Letter 0]

24 Oct 2024

Point-to-Point Responses to the Reviewers’ Critiques (PONE-D-24-23687)

Notes. For purposes of this response letter, we have created figures to help illustrate our points. These figures do not appear in the manuscript.

For Journal Requirements:

1. When submitting your revision, we need you to address these additional requirements. Please ensure that your manuscript meets PLOS ONE's style requirements, including those for file naming.

Response: We thank the editor for suggestting this very important requirements, we will vevise the style of our manuscript to meet PLOS ONE’s style. 

2. We noticed you have some minor occurrence of overlapping text with previous publication(s), which needs to be addressed.

 Response: We thank the editor for pointing out this mistake. We have already polished the language and reduced the repetition in the manuscript and ensure that in our resubmitted manuscript this problem will be addressed.

Response: We appreciate the editor for highlighting this mistake. The issue with the mismatched grant information has been resolved.

Response: We appreciate the editor for comment. We have updated the author information and added ORCID of Prof. Qin.

Reviewer #1:

1. The study utilized data from multiple databases, yet it is still necessary to clarify whether the sample selection is representative.

Response: 

We appreciate the reviewer’s insightful comment and fully agree on the importance of representative sample selection for our study. To ensure this, we integrated data from multiple databases. We obtained extensive RNA-seq and clinical information from The Cancer Genome Atlas (TCGA) and Genotype-Tissue Expression (GTEX) databases, covering 33 tumor types and their corresponding normal tissue samples. This comprehensive data source ensures a wide coverage of cancer types and includes sufficient normal tissue controls, thereby enhancing the universality and reliability of our findings.

Additionally, we utilized tumor cell line data from the Cancer Cell Line Encyclopedia (CCLE) database to further increase sample diversity. To investigate the role of B3GNT5 in tumor development and immunotherapy, we conducted genetic variation analyses, including mutation status, gene copy number, and methylation levels. This multi-dimensional analysis helps us fully understand the expression patterns and potential mechanisms of B3GNT5 in various cancers.

We also employed Kaplan-Meier survival analysis and the Cox regression model to assess the impact of B3GNT5 on patient overall survival. These methods consider time factors and clinical outcomes, lending stability to our conclusions. Moreover, similar high-impact pan-cancer studies have used the same databases (1-3) (the paper we listed below), indicating the completeness of our data, which includes genome sequencing and postoperative follow-up survival data. In cases of incomplete data, we opted for exclusion.

In summary, this study effectively ensures the representativeness and scientific validity of sample selection through cross-database data integration, multi-faceted verification, and robust statistical methods. This systematic research design provides a solid foundation for elucidating the role of B3GNT5 in cancer.

2. When conducting gene set enrichment analysis and gene set variation analysis, multiple comparisons might be involved, but the text does not mention whether multiple comparison corrections were applied.

Response: We thank the reviewer for this insightful comment. In the gene set enrichment analysis and gene set variation analysis, multiple comparisons are indeed made, and we need to adjust our P-value to control the error rate, and the adjusted P-value is the original P-value adjusted by Bonferroni or Benjamini-Hochberg method

3. In the part "Regulating the expression of B3GNT5 can affect self-renewal of PAAD cells", "NC" is mentioned, but the specific setup of the NC group and the type of control are not detailed. In addition, it is not specified whether the experiment was repeated, the number of repetitions, or whether a control group was set up.

Response: We thank the reviewer for this insightful comment. We completely agree with your opinion that setting up a control group and repeating the experiment is of utmost significance. In our study of regulating the expression of B3GNT5 to affect the self-renewal of pancreatic adenocarcinoma (PAAD) cells, NC means negative control group, it refers to the cell group transfected with empty plasmid. This setting helps to rule out the influence of non-specific effects, thus verifying that the changes observed in the experimental group are indeed caused by the changes in B3GNT5 expression levels.

Reviewer #2:

1. Please ensure that page and line numbers are provided in the correct format.

Response: Thank you very much for pointing out this detail. We have revised it. 

2. Please note that the current lack of clarity of the figures makes them more difficult to read. The figures have to be prepared at high resolution in a clear and meaningful manner. Additionally, providing a more detailed legend would help readers understand the data presented.

Response: Thank you for pointing out this mistake. While we revise the manuscript, we’ll upload the high-resolution figures with a minimum resolution of 300dpi. In the meantime, we will provide more detail in figure legend that could help readers understand data presented.

3. The font formatting of the references needs to be adjusted. Font style and size should be uniform throughout the manuscript, including references.

Response: Thank you very much for your suggestion. We have improved the manuscript format according to the journal style. We hope the revised version is more in line with requirements.

4. The introduction provides a good background on B3GNT5; however, it could benefit from a more detailed discussion on the specific mechanisms by which B3GNT5 influences immune responses in tumors.

Response: Thank you very much for your suggestion. We have added a more detailed discussion on the specific mechanisms by which B3GNT5 influences immune responses in tumors. The B3GNT5 affects tumor immunity by regulating the synthesis of glycolipids (GSL) on the cell surface, participating in the process of epithelial-mesenchymal transition (EMT), and promoting the secretion of TGF-β and CCL2 factors by CSCs (cancer stem cells)(4, 5).

5. It would be helpful to include more details on the statistical analyses performed, including any software used and specific tests applied.

Response: Thank you very much for your suggestion. In the process of data analysis, we use a variety of bioinformatics tools and software, such as the survival package in R language for survival analysis, and use GraphPad Prism software to complete the chart making. At the same time, in order to evaluate the degree of immune cell infiltration, we also visited the data provided by TCGA and GEO projects and analyzed them using a specific online platform (http://cibersortx.standford.edu/)

6. The discussion effectively contextualizes the findings within the broader field of cancer research. Are there specific experimental approaches or clinical trials that could be proposed to confirm the role of B3GNT5 in cancer immunotherapy? This would strengthen the manuscript by providing a pathway for future research cancer stem cell stemness.

Response: We thank the reviewer for this insightful comment. Regarding the role of B3GNT5 in cancer immunotherapy, the existing research mainly focuses on the relationship between its expression and immune cell infiltration in the tumor microenvironment (TME). Through gene expression analysis, it was found that the expression level of B3GNT5 was negatively correlated with the number of CD8 + T cells, CD4 + T cells and NK cells. In addition, in pancreatic cancer cell line PANC-1, inhibiting the expression of B3GNT5 can significantly reduce the ability of cell sphere formation, suggesting that B3GNT5 may be one of the potential immunotherapy targets. In future studies, we will further verify the specific role of B3GNT5 in cancer immunotherapy. For example, CRISPR/Cas9 technology was used to accurately edit B3GNT5 gene in human tumor cell line, and then co-cultured with different types of immune cells to detect the changes of immune cell activity and cytokine secretion. The mouse tumor model with B3GNT5 knockout or overexpression was constructed, and the tumor growth and the response difference to immune checkpoint inhibitors (such as PD-1/PD-L1 antibody) were observed. This is helpful to evaluate the role of B3GNT5 in immune escape mechanism. Based on the above experimental results, a suitable animal model was selected for pre-clinical test to evaluate the safety and effectiveness of combined use of B3GNT5 inhibitor and existing immunotherapy methods. Through the above research, we can not only deeply understand the function of B3GNT5 in the process of tumor occurrence and development, but also provide scientific basis for it as a new immunotherapy target. More large-scale clinical trials are needed in the future to verify these findings and explore the best treatment strategies.

7. The authors need to improve their language and grammar to enhance the flow of the text. For example, the phrase "B3GNT5's expression was highly correlated with different immunoregulatory factors" could be rephrased to "The expression of B3GNT5 shows a high correlation with a range of immunoregulatory factors." This change would enhance the clarity of the statement. In addition, the whole manuscript needs to be checked by native English speakers.

Response: Thank you for your comments. We have found a professional editing agency, to help us modify the English language. We hope the revised version will be of much better reading quality.

Reviewer #3:

1. The article should undergo extensive English proofreading.

Response: Thank you for your comments. We have engaged a professional editing agency to assist with refining the English language. We hope the revised version will offer significantly improved readability.

2. Too many full stops. They need to be removed. For example: This phenomenon is particularly visible in underdeveloped countries, where about 82% of the world's population lives.

Response: Thank you very much for pointing out this mistake. We have revised it.

3. In the methodology section: transfection is not sufficiently described. please provide details.

Response: Thank you very much for your suggestion, we will provide more details about transfection in the methodology section.

4. In the results section, each paragraph should include a sentence summarizing what the results actually mean.

Response: Thank you for this insightful comment. We have added a summary sentence to each paragraph to ensure that readers can clearly understand the implications of the results.

5. There are no results confirming that sh-RNA transfection was successful. Please analyze B3GNT5 expression after shRNA silencing. Without confirmation of inhibition, it is not possible to show these results and draw any conclusions.

Response: Thank you for your valuable comment. We conducted an examination to analyze B3GNT5 expression following shRNA silencing. Using qRT-PCR, we assessed B3GNT5 expression in Panc-1 cells transfected with sh-B3GNT5 and a negative control. The figure below illustrates the relative expression of B3GNT5 in Panc-1 cells transfected with sh-B3GNT5 compared to those transfected with the negative control.

6. The resolution of the figures is too small and the results are blurry.

Response: Thank you for highlighting this mistake. As we revise the manuscript, we will upload the high-resolution figures with a minimum resolution of 300 dpi. Additionally, we will enhance the figure legends to provide more detail, aiding readers in understanding the presented data.

1. Wu Z, Uhl B, Gires O, Reichel CA. A transcriptomic pan-cancer signature for survival prognostication and prediction of immunotherapy response based on endothelial senescence. J Biomed Sci. 2023;30(1):21.

2. Zhuang K, Wang L, Lu C, Liu Z, Yang D, Zhong H, et al. Assessment of SWI/SNF chromatin remodeling complex related genes as potential biomarkers and therapeutic targets in pan-cancer. Mol Cancer. 2024;23(1):176.

3. Luo H, Xia X, Huang LB, An H, Cao M, Kim GD, et al. Pan-cancer single-cell analysis reveals the heterogeneity and plasticity of cancer-associated fibroblasts in the tumor microenvironment. Nat Commun. 2022;13(1):6619.

4. Miao Z, Cao Q, Liao R, Chen X, Li X, Bai L, et al. Elevated transcription and glycosylation of B3GNT5 promotes breast cancer aggressiveness. J Exp Clin Cancer Res. 2022;41(1):169.

5. Zhang X, Zeng B, Zhu H, Ma R, Yuan P, Chen Z, et al. Role of glycosphingolipid biosynthesis coregulators in malignant progression of thymoma. Int J Biol Sci. 2023;19(14):4442-56.

---

## [Decision Letter · Decision Letter 1]

14 Nov 2024

Pan-cancer Analysis of B3GNT5 with Potential Implications for Cancer Immunotherapy and Cancer Stem Cell stemness

PONE-D-24-23687R1

Dear Dr. Qin,

We’re pleased to inform you that your manuscript has been judged scientifically suitable for publication and will be formally accepted for publication once it meets all outstanding technical requirements.

Kind regards,

Zhiwen Luo

Academic Editor

PLOS ONE

Additional Editor Comments (optional):

Reviewers' comments:

Reviewer's Responses to Questions

**Comments to the Author**

1. If the authors have adequately addressed your comments raised in a previous round of review and you feel that this manuscript is now acceptable for publication, you may indicate that here to bypass the “Comments to the Author” section, enter your conflict of interest statement in the “Confidential to Editor” section, and submit your "Accept" recommendation.

Reviewer #3: All comments have been addressed

2. Is the manuscript technically sound, and do the data support the conclusions?

Reviewer #3: Yes

3. Has the statistical analysis been performed appropriately and rigorously? 

Reviewer #3: Yes

4. Have the authors made all data underlying the findings in their manuscript fully available?

Reviewer #3: Yes

5. Is the manuscript presented in an intelligible fashion and written in standard English?

Reviewer #3: Yes

6. Review Comments to the Author

Reviewer #3: The authors took into account all my comments, hence in my opinion the article meets the requirements for publication.

7. PLOS authors have the option to publish the peer review history of their article (what does this mean?). If published, this will include your full peer review and any attached files.

Reviewer #3: No

---

## [Editor Report · Acceptance letter]

3 Dec 2024

PONE-D-24-23687R1 

PLOS ONE

Dear Dr. Qin, 

I'm pleased to inform you that your manuscript has been deemed suitable for publication in PLOS ONE. Congratulations! Your manuscript is now being handed over to our production team.

Kind regards, 

on behalf of

Dr. Zhiwen Luo 

Academic Editor

PLOS ONE